# Characterisation of the British honey bee metagenome

Tim Regan [1], Mark W. Barnett[1], Dominik R. Laetsch [2], Stephen J. Bush[1], David Wragg[1], Giles E. Budge[3,4], Fiona Highet[5], Benjamin Dainat[6], Joachim R. de Miranda [7], Mick Watson [1], Mark Blaxter[2] & Tom C. Freeman[1]

The European honey bee (Apis mellifera) plays a major role in pollination and food production. Honey bee health is a complex product of the environment, host genetics and associated microbes (commensal, opportunistic and pathogenic). Improved understanding of these factors will help manage modern challenges to bee health. Here we used DNA sequencing to characterise the genomes and metagenomes of 19 honey bee colonies from across Britain. Low heterozygosity was observed in many Scottish colonies which had high similarity to the native dark bee. Colonies exhibited high diversity in composition and relative abundance of individual microbiome taxa. Most non-bee sequences were derived from known honey bee commensal bacteria or pathogens. However, DNA was also detected from additional fungal, protozoan and metazoan species. To classify cobionts lacking genomic information, we developed a novel network analysis approach for clustering orphan DNA contigs. Our analyses shed light on microbial communities associated with honey bees and demonstrate the power of high-throughput, directed metagenomics for identifying novel biological threats in agroecosystems.

[1] The Roslin Institute and Royal (Dick) School of Veterinary Studies, University of Edinburgh, Easter Bush EH25 9RG Edinburgh, UK. [2] The Institute of Evolutionary Biology, School of Biological Sciences, The University of Edinburgh, Edinburgh EH9 3JG, UK. [3] Fera, The National Agrifood Innovation Campus, Sand Hutton YO41 1LZ York, UK. [4] School of Natural and Environmental Sciences, Newcastle University, Newcastle upon Tyne NE1 7RU, UK. [5] Science and Advice for Scottish Agriculture, 1 Roddinglaw Road, Edinburgh EH12 9FJ, UK. [6] Agroscope, Swiss Bee Research Centre, Schwarzenburgstrasse 161, CH-3003 Bern, Switzerland. [7] Department of Ecology, Swedish University of Agricultural Sciences, Uppsala 750 07, Sweden. These authors contributed equally: Tim Regan, Mark W. Barnett. Correspondence and requests for materials should be addressed to T.R. (email: tim.regan@roslin.ed.ac.uk) or to M.B. (email: mark.blaxter@ed.ac.uk) or to T.C.F. (email: tfreeman@roslin.ed.ac.uk)

The European honey bee, *Apis mellifera* Linnaeus, has a global distribution and a major role in pollination and food production[1]. Like other pollinators, honey bee populations face multiple threats. There is increasing evidence of pollinator decline globally. Whilst flowering crops benefit greatly from a diversity of insect pollinators[2], managed honey bees are a major global contributor, providing nearly half of the service to all insect-pollinated crops on Earth[3,4]. Despite the recent increase in non-commercial beekeeping, the number of managed honey bee colonies is growing more slowly than agricultural demand for pollination[5]. The decline in pollinators is not thought to be caused by a single factor but may be driven by a combination of habitat fragmentation, agricultural intensification, pesticide residue accumulation, new honey bee pests and diseases, and suboptimal beekeeping practices[6–8]. Trade in honey bees from different regions of the globe have unquestionably contributed to a rise in infectious disease and there may be transmission between honey bees and wild pollinators[9–11].

The genetic structure of British honey bee populations has undergone large changes over the last 100 years. The native M-lineage subspecies, *A. m. mellifera*, had predominated in Britain, but the population was decimated in the early 20th century by a combination of poor weather and chronic bee paralysis virus, thought to have been the cause of Isle of Wight disease[12]. Following this, the practice of bee importation increased dramatically. In Britain today there is a growing industry that imports bees from mainland Europe, particularly the Italian honey bee (*A. m. ligustica*) and Carniolan honey bee (*A. m. carnica*), both C-lineage subspecies. Importation of queens has for a long time been used as a means to compensate for the loss of colonies and the Southern European strains are often viewed as a means to improve honey production. It had been assumed that the native British bee was extinct, but new molecular studies have shown that colonies robustly assigned to *A. m. mellifera* still exist in Northern Europe[13,14]. The genetic diversity of British honey bee populations is poorly understood. The genetic makeup of bee populations not only influences production traits and the ability to survive under less favourable conditions, but also plays a vital role in disease resistance[15].

The health of British honey bees is under threat from a range of native and non-native bacterial, fungal and viral pathogens. While known 'notifiable diseases' can be risk assessed and regulated by law, emergent diseases such as *Nosema ceranae*[16] may be spread globally before they have been properly identified and risk assessed. Nosemosis is one of the most prevalent honey bee diseases and is caused by two species of microsporidia, *Nosema apis* and *Nosema ceranae*, that parasitise the ventriculum (midgut). Although infected bees often show no clear symptoms, heavy infections can result in a broad range of detrimental effects[17–22]. *N. ceranae*, a native parasite of the Asiatic honey bee (*Apis cerana*), has been detected in *Apis mellifera* samples from Uruguay predating 1990 but is now present in *Apis mellifera* worldwide[16]. Notifiable diseases, American foulbrood (AFB) and European foulbrood (EFB), are caused by the bacteria *Paenibacillus larvae* and *Melissococcus plutonius*, respectively[23,24]. Acarine disease is caused by a mite found throughout Britain which infests the trachea of honey bees[25]. Protozoans such as gregarines and the emergent trypanosomatid *Lotmaria passim*, also infect honey bees. The most devastating of all introduced pathogenic species in recent years is the hemophagous mite *Varroa destructor*, which shifted hosts from *A. cerana* to *A. mellifera* sometime in the first half of the 20th century[26]. *Varroa* mites feed on the haemolymph of both larval and adult stages of the honey bee. More importantly, *V. destructor* transmits several bee viruses, generating epidemics that kill colonies within 2–3 years unless the *Varroa* population is kept under control. Among the most important and lethal viruses in this regard are deformed wing virus (DWV)[27], acute bee paralysis virus complex (ABPV), Kashmir bee virus (KBV), and Israeli acute paralysis virus (IAPV)[28]. Sacbrood virus (SBV) can also be transmitted but without major epidemic consequences and is primarily indirectly affected by *Varroa*[26,29,30].

In several species, the core commensal microbiome can mediate disease susceptibility and the internal ecology of the host can greatly affect disease outcome[31], e.g. bumblebee gut microbiota composition has a stronger effect on susceptibility to the parasite *Crithidia bombi* than host genotype[32]. In addition to immunological health and essential nutrient provision, microbial metabolism affects the growth, behaviour and hormonal signalling of honey bees[33]. Unlike most host species, the core microbiota of the honey bee has relatively little diversity[34–40]. *Snodgrasella alvi* (Betaproteobacteria), *Gilliamella apicola* (Gammaproteobacteria), two *Lactobacillus* taxa (Firm-4 and Firm-5)[36,37], and *Bifidobacterium asteroides* are common and abundant[41,42]. There are at least four less common species: *Frischella perrara*[43], *Bartonella apis*[44], *Parasaccharibacter apium*[39] and Gluconobacter-related species group Alpha2.1[37]. Metagenomic analyses have revealed high between-isolate genetic diversity in honey bee microbiotal taxa, suggesting they comprise clusters of related taxa[45]. These bacteria maintain gut physiochemical conditions and aid their host in the digestion and metabolism of nutrients, neutralisation of toxins, and resistance to parasites[40,46,47]. *Gilliamella* species digest pectin from pollen, and the *Lactobacillus* species inhibit the growth of foulbrood bacteria[48]. However, *F. perrara* may cause a widespread scab phenotype in the gut[49]. A negative correlation was found between the presence of *Snodgrasella alvi* and pathogenic *Crithidia* in bees[50], but pre-treatment of honey bees with *S. alvi* prior to challenge with *Lotmaria passim* (an *A. mellifera* pathogen closely related to *Crithidia*) resulted in greater levels of *L. passim* compared to bees which were not pre-treated[51]. Thus, commensal microbiome species can have beneficial, mutual or parasitic relationships with their hosts, and in particular, different combinations of species – different microbiota communities – may be associated with variations in honey bee health.

With recent significant reductions in the cost of high throughput sequencing, metagenomics could be a useful tool for analysing genetic lineage, gut health and pathogen load as part of routine testing and/or monitoring imports for novel pathogens. Here, to establish baseline figures and test the suitability of this approach, we applied deep sequencing of the honey bee metagenome together with a novel network analysis framework, to examine the genomes of honey bees and their symbiotic and pathogenic cobionts in British apiaries.

## Results

**Metagenome sequencing of honey bees and their cobionts**. We performed full metagenomic sequencing of 19 samples of British honey bees (Supplementary Table 1). Samples were obtained from hives located across Scotland and England (Fig. 1a), each sample comprised of 16 workers collected from a single colony. Duplicates of samples 1–4 were analysed at a lower sequencing coverage to assess cobiont and genomic variant discovery (samples 8–11). While the sample size was limited, the colonies sequenced were selected so as to represent bees from diverse geographical locations and to be representative of the phenotypic diversity of honey bees currently managed by British apiarists. Notably, representatives of the Buckfast bee and the Colonsay "native" black bee lines were included in the sampling. The entire thorax and abdomen was processed for genome sequencing, thus including gut microorganisms, organisms attached to the outside of the bees, and haemolymph/tissue parasites. Between 15 and 45 million 125 base paired-end reads were generated per sample on

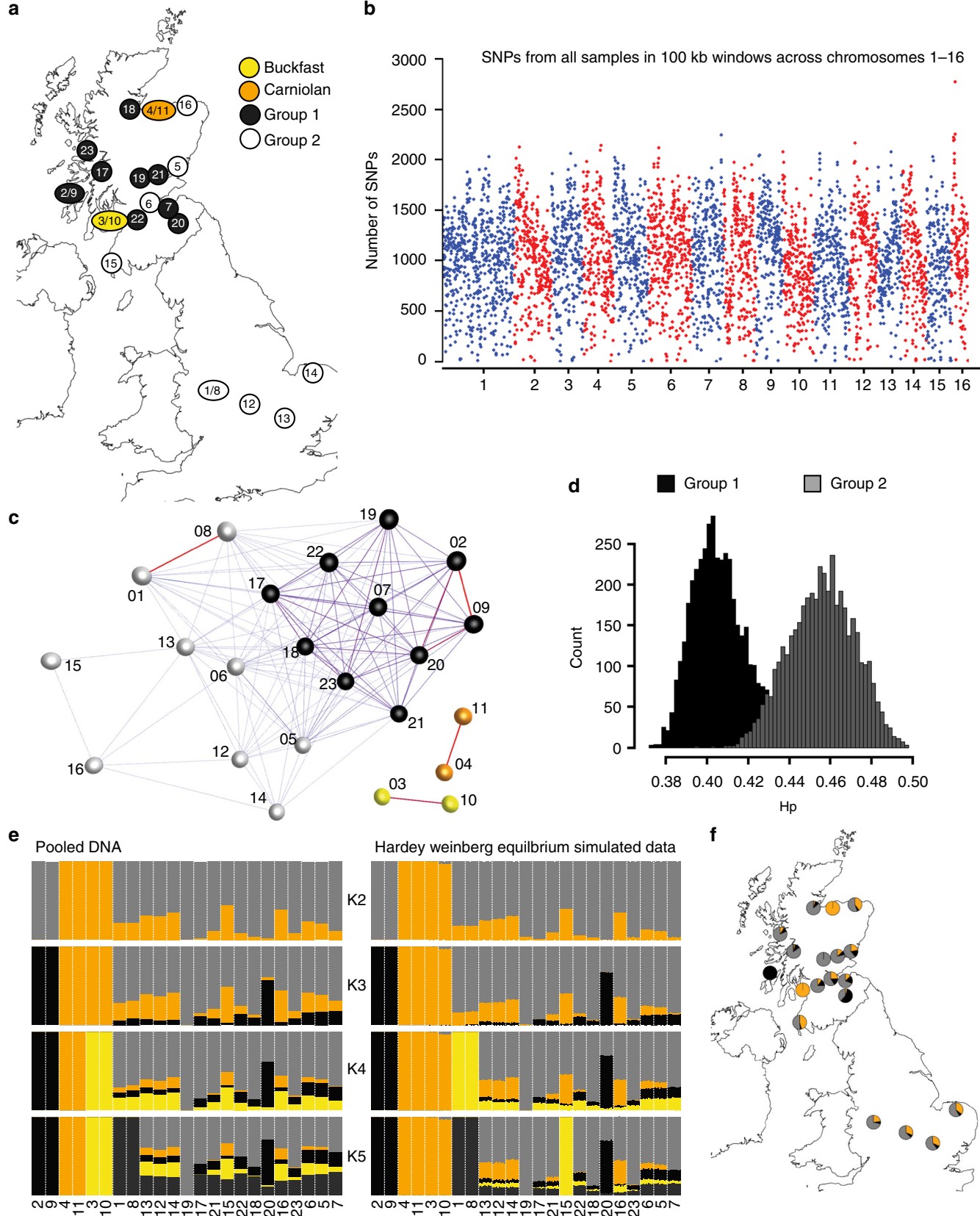

the Illumina HiSeq 2500, equivalent to between 17- and 50-fold coverage of the honey bee genome (Amel 4.5).

**Genomic diversity of sampled honey bees.** DNA sequence data were mapped onto the honey bee reference genome (version Amel

4.5[52]) and variants identified. Overall 3,940,467 sites were called as polymorphic, ranging from 962,775 to 2,586,224 single nucleotide variants (SNVs) per sample (Fig. 1b). A network graph derived from a matrix of identity-by-state (IBS) at each variant position for all samples was used to define related groups of samples (Fig. 1c).

**Fig. 1** *Apis mellifera* diversity. **a** A map of the UK with the location of colonies sampled. **b** The number of SNVs from all samples presented across *A. mellifera* chromosomes 1 to 16 in 100 kb consecutive windows. **c** A network based on the identity by state (IBS) similarity score of sample variants identifying Group 1 in the centre and Group 2 in the periphery of the major cluster while Carniolan and Buckfast samples remain distinct. This includes sequencing duplicates (01–04 and 08–11). Strength of edges is represented on a scale from thin and blue (weak) to thick and red (strong). **d** The heterozygosity level across consecutive window of size 100 kb comparing groups 1 and 2 identified from the network graph. **e** ADMIXTURE analyses of pooled DNA (left) and genotypes simulated assuming Hardy Weinberg equilibrium (right); colours indicate the distinct genetic backgrounds identified assuming K backgrounds. **f** Map of sampling locations indicating ADMIXTURE results at $K = 3$. Maps were obtained from © EuroGeographics. Original product is available for free at www.eurogeographics.org Terms of licence available at https://eurogeographics.org/services/open-data/topographic-data/

Group 1, which includes the native black bee sample from Colonsay (samples 2 and 9), was less heterozygous than Group 2 (Fig. 1d). ADMIXTURE[53] analyses were used to explore population subdivision in the data following removal of SNVs in linkage disequilibrium. ADMIXTURE cross-validation (CV) error values increased as the number of populations (K) assumed to be contributing to the variation were increased ($K = 1$, CV = 0.562; $K = 2$, CV = 0.601; $K = 3$, CV = 0.712; $K = 4$, CV = 0.853; $K = 5$, CV = 1.007). At $K = 2$ the Buckfast (samples 3 and 10) and Carniolan (samples 4 and 11) C lineage samples were distinguished from the M lineage *A. m. mellifera* samples, while $K = 3$ further discerns the "native" *A. m. mellifera* sampled from Colonsay (samples 2 and 9), the Buckfast sample at $K = 4$ and the *A. m. mellifera* breeding project (samples 1 and 8) at $K = 5$ (Fig. 1e).

ADMIXTURE was originally designed to estimate ancestry in unrelated individuals rather than pooled DNA from several individuals, as analysed here. To address this, genotypes were simulated for 10 individuals per pooled DNA sample, using allele sequence depth to estimate allele frequency under an assumption of Hardy–Weinberg equilibrium and analysed using ADMIXTURE. The CV error values decreased as K was increased ($K = 1$, CV = 0.980; $K = 2$, CV = 0.835; $K = 3$, CV = 0.795; $K = 4$, CV = 0.763; $K = 5$, CV = 0.736). At $K \leq 3$ the simulated data results were consistent with those from the actual pooled genotypes, while $K = 4$ distinguished samples from the *A. m. mellifera* breeding project (samples 1 and 8), and $K = 5$ assigned a distinct genetic background to bees sampled from Wigtownshire (sample 15) (Fig. 1e). k-nearest neighbour (kNN) network analysis of the pooled genotype data using NetView[54,55] also identified 2 clusters, separating C and M lineage samples in the same manner as the ADMIXTURE analyses (Supplementary Fig. 1). Together, these results support a model of two genetic backgrounds in the British bee populations sampled, most likely representing the C and M lineages, with evidence of a distinct *A. m. mellifera* background in bees originating from Colonsay and other areas of Scotland, and differentiation of Buckfast and Carniolan bees (Fig. 1f).

**The microbiome of honey bees.** The majority of the data (~90% of reads) from each sample mapped to the honey bee reference genome. Reads that did not map to the honey bee reference were collated and used for a metagenomic assembly. This resulted in over 35,000 contigs greater than 1 kb in length. Contigs were assigned to a taxonomic group by comparison to a series of curated databases in a defined order (Fig. 2a) using BlobTools[56]. First, contigs were compared to the bee cobiont sequence data in the HoloBee Database (v2016.1)[57], followed by genomes and proteomes of species identified as being bee-associated[58,59], and finally by comparison of contigs against the NCBI Nucleotide and UniProt Reference Proteome databases. Patterns of coverage, GC% and taxonomic annotation of contigs were explored to identify likely genomic compartments present (Fig. 2b, c). We

discarded contigs with read coverage lower than 1, as these were likely an artefact of pooling reads, yielding a final set of 31,386 metagenome contigs, spanning 140 Mb. Taxon assignments are summarised in Supplementary Table 2. Correlation graphs were generated in order to examine: (1) how similar bees were based on the overall composition of their microbiome; (2) to group contigs based on their relative abundance across samples. Clustering samples based on the composition of their microbiome did not recapitulate their clustering by honey bee genome SNVs (Fig. 2d). A graph was also constructed where nodes represented individual contigs and the relationships between them (edges), were defined by the correlation between their abundance profiles (base coverage) across samples (Fig. 3). A high correlation threshold ($r = 0.99$) was used, to minimise spurious correlations, although ~35% of the contigs were unconnected and do appear in the graph. The highly structured multi-component graph was subdivided using the MCL algorithm[60] into clusters of contigs whose abundance across the samples was very similar. Many of these clusters were made up of contigs derived from the same species or in a number of cases from strongly co-occurring species.

Rarefaction analysis of ribosomal RNA sequences present in the assembled data was used to estimate the species richness discovered as a function of sequencing depth (Supplementary Fig. 2). While there was variation between samples in terms of species richness at all sequencing depths, even the lowest coverage achieved (17x reference genome coverage) was likely to be sufficient to capture most *A. mellifera* cobionts present, as samples with higher coverage contained few new cobionts over samples at lower coverage.

We examined graph clusters further. One (Fig. 4a) contained 1.33 Mb of sequence, most of which had no match in public databases, but contained some contigs that had significant similarity to sequences from other *Apis* species (Fig. 4b). The number of reads mapping to these contigs was proportional to the depth of sequencing (Fig. 4c) and we infer that they likely represent contigs from the *A. mellifera* genome not present in the honey bee reference genome (Fig. 4d). Others in this cluster, spanning 0.01 Mb, matched sequences from *Ascophaera apis* (chalkbrood), an endemic fungal associate of honey bees[61].

Most of the other groups of contigs could be assigned to cobiont organisms. The contribution of non-*A. mellifera* reads varied between samples, a pattern that may be partly explained by the presence in some samples of eukaryotic pathogens such as *Nosema* microsporidians and the trypanosomatid *L. passim*, which have larger genomes. The most abundant non-pathogenic bacterial cobionts identified were *Gilliamella apicola*, *Bartonella apis*, *Frischella perrara*, *Snodgrassella alvi*, "Firm-4" firmicutes[53] (*Lactobacillus mellis* and *Lactobacillus mellifer*), "Firm-5" firmicutes[53] (*Lactobacillus melliventris*, *Lactobacillus kimbladii*, *Lactobacillus kullabergensis*, *Lactobacillus sp.* wkB8, *Lactobacillus helsingborgensis* and *Lactobacillus sp.* wkB10), *Lactobacillus kunkeei* and *Bifidobacterium asteroides* (Supplementary Table 2). Each species varied in its abundance across the samples. In some

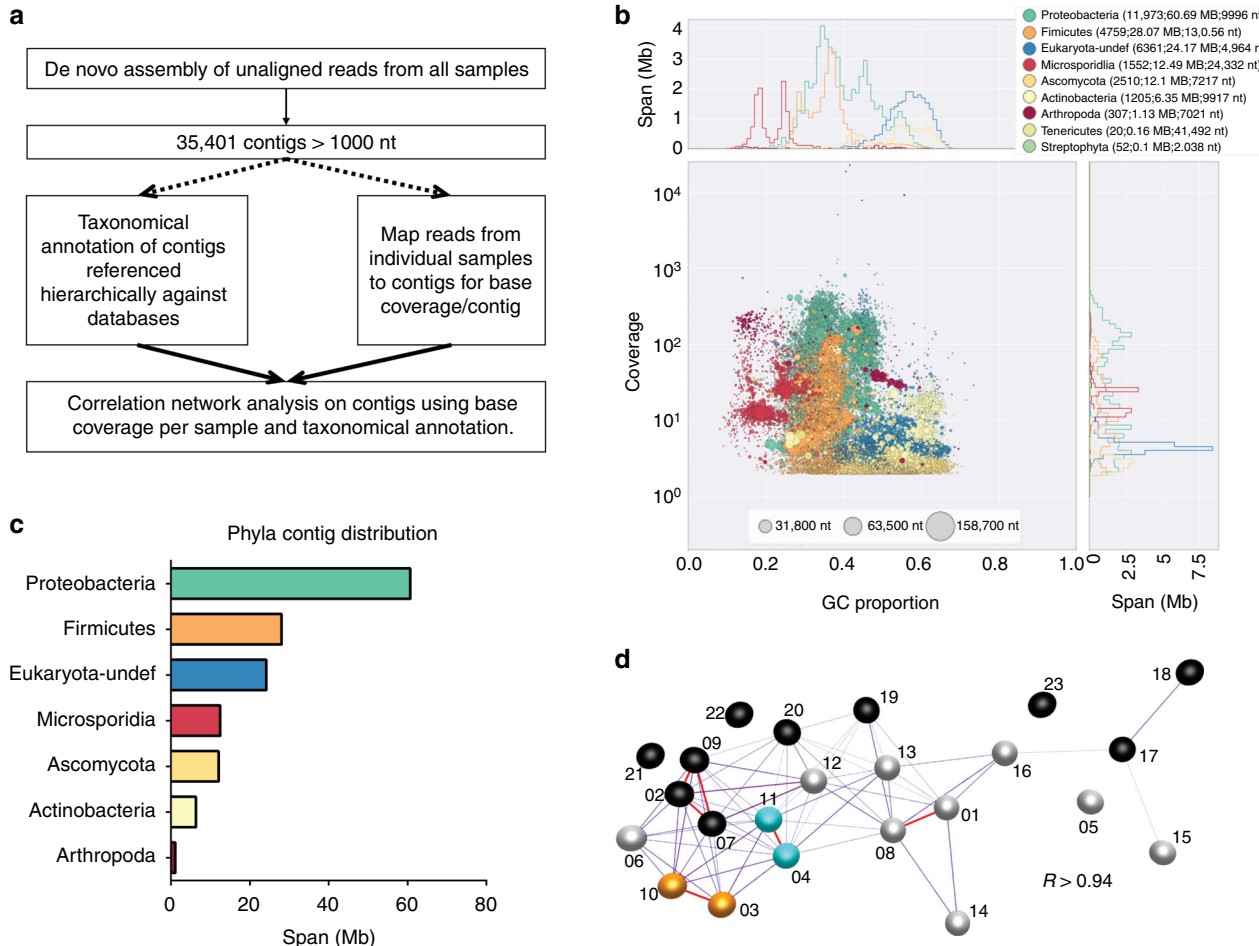

**Fig. 2** Metagenomics of *Apis mellifera*. **a** A flow diagram of the microbiome analysis using reads which did not align to the *Apis mellifera* reference genome. **b** A blobplot generated from contigs using unaligned reads from all samples. Contigs are plotted based on their GC content (*x*-axis) and coverage (*y*-axis), scaled by span, and coloured by their phylum assignation. **c** The span of de novo assembled contigs which were assigned to given phyla is displayed for the 12 most abundant phyla across all samples. **d** A network based on the coverage/contig from each sample representing microbiome composition/unaligned reads

nominal species, contig clustering suggested the presence of multiple distinct genotypes of cobionts. For example, contigs ascribed to *Bartonella apis* together had a total span of 11.7 Mb, almost five times longer than the reference *B. apis* genome, and formed a connected network module (Fig. 5a). The three largest *B. apis* clusters had distinct distribution across the samples, which likely reflects the presence of distinct genotypes of *B. apis* with varying abundance across the samples. Similarly, contigs ascribed to *Gilliamella apicola*, the most abundant species identified in the bee microbiome, were distributed across a number of clusters with related but different abundance profiles (Fig. 5b). Clusters containing contigs from several closely related but distinct *Lactobacillus* species were identified: Firm-4 lactobacilli (clusters 25 and 40) or Firm-5 lactobacilli (clusters 16, 20, 21 and 24) (Fig. 5c). These *Lactobacillus* groups may represent a distinct cobiont community whose abundance is linked, but sufficiently different to allow separation of their contigs. The exception was cluster 21, which contained contigs assigned to a mix of Firm-5 species: this may represent a core genome component conserved between species. Cluster 29 comprised contigs assigned to *Lactobacillus kunkeei* that formed an unconnected graph component. *L. kunkeei* is thought to be an environmental rather than a gut microbiome organism. Some connected components were more complex. Cluster 32 contained contigs assigned to

several prevalent honey bee cobionts, including *G. apicola*, *F. perrara*, *B. asteroides*, *S. alvi*, *B. apis*, *S. floricola* and *P. apium*. The co-clustering of genomic segments from multiple species is likely to reflect a strongly interacting community of organisms where the relative abundance of each is regulated homeostatically[45,59,62].

Some clusters had very restricted presence in the sample set. For example, cluster 3 was largely restricted to sample 4 (Supplementary Fig. 3e). These are likely to derive either from rare members of the honey bee cobiont community or opportunistic infections. Several clusters had little to no annotation (Supplementary Fig. 3f). The coverage of these contigs was also usually derived from individual samples. They may represent novel species, or divergent or novel genomic regions of known species.

**Honey bee pathogens.** Known honey bee pathogens were detected in many samples. One of the largest components of clustered contigs was assigned to the trypanosomatid parasite *Lotmaria passim*, with a combined span of 16.3 Mb (Fig. 6a). While sequences were detected from notifiable pathogens *Melisococcus plutonious* and *Paenibacillus larvae* (European and American foulbrood), no distinct cluster was identified and the <1 Mb total combined span of matched sequences was relatively minor (Supplementary Table 2).

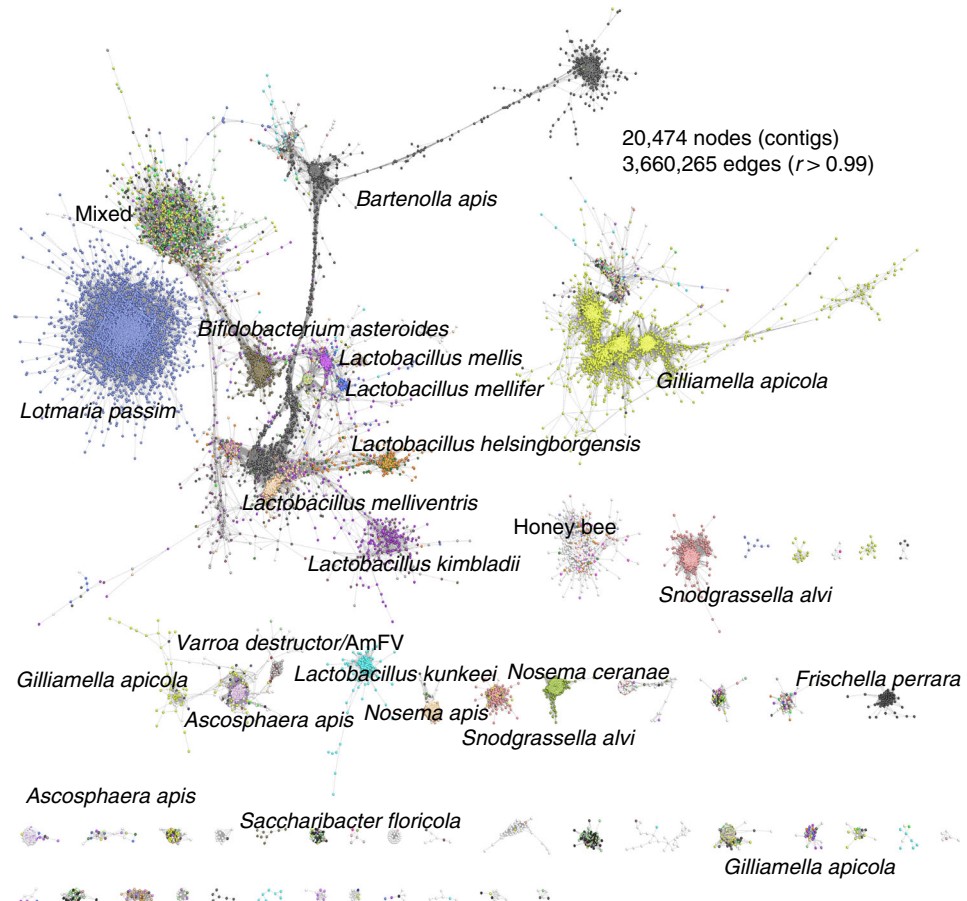

**Fig. 3** Correlation network analysis of microbiome contigs. Each node represents an individual contig and edges are defined based on the correlation between abundance profiles (base coverage) across individual samples. Contigs (nodes) are connected if the Pearson correlation between two contigs abundance profile was $r > 0.99$. Each contig is coloured according to the species it maps to, white nodes represent contigs for which no significant sequence match was found

Both *Nosema* species *N. apis* (Fig. 6b) and *N. ceranae* (Fig. 6c) were identified. *N. ceranae* was more prevalent (5/19 colonies vs. 2/19 colonies). Contigs matching the pathogen causing "chalk brood" (*Ascophaera apis*) were found in cluster 2 and were derived almost exclusively from sample 23 (Fig. 6d). Cluster 47 contained contigs assigned to the parasitic mite *V. destructor* and contigs assigned to *Apis mellifera filamentous virus* (AmFV), found in 6/19 colonies (Fig. 6c). The largest source of reads mapping to these contigs was sample 23, which also had a high prevalence of chalkbrood. Blobplots describing the taxonomy and cumulative span for each panel in Fig. 6 are available in Supplementary Fig. 3d–j.

The 'completeness' of the metagenomic assemblies was analysed for each of the clusters using checkM and compared to the metagenomic binning as performed by MetaBAT[63]. MetaBAT uses both coverage information and sequence context (tetranucleotide frequencies) to bin genomes, while our network clustering relied on coverage information alone. CheckM uses a set of pre-computed core genes to assess the completeness and contamination. MetaBAT yielded eighteen bacterial genome assemblies at >80% complete compared totwenty assemblies using our method. Results are displayed in Supplementary Tables 4 and 5. CheckM also attempts to assign a taxonomic level to each metagenome assembled genome, but is not appropriate for eukaryotic genomes. For this, we used Benchmarking Universal Single-Copy Orthologs (BUSCO)[64] to analyse the clusters associated with *Ascophaera apis*, *Lotmaria passim*, *Nosema apis*, *Nosema ceranae* and *Varroa destructor* genomes (Supplementary Figure 4).

To validate the metagenomic hits, we employed PCR to screen our samples for *B. apis, Nosema ceranae* and *L. passim*. All samples in which we identified sequences deriving from these organisms were positive by PCR. However, we also identified the presence of species in additional samples not scored as positive by sequencing, suggesting that the PCR assays are more sensitive than bulk sequencing (Supplementary Fig. 5a–c). We also identified a small cluster containing only one contig matching to a recorded genome sequence, *Apicystis bombi*, a gregarine known to parasitise honey bees[65]. To identify the exact species present, we sequenced the PCR results of custom primers against the largest contig in this cluster, in conjunction with primers encompassing the 18 S and ITS2 rDNA regions, as used by Dias et al. for the characterisation of novel gregarine species[66] (Supplementary Fig. 5d). The contig sequence matched various gregarine species, while the ribosomal DNA sequence confirmed the species present to be *Apicystis bombi* (Supplementary Fig. 5e).

## Discussion

A healthy population of honey bees is crucial for the security of the ecosystem service of pollination. With the continued and sometimes unregulated global transport of *A. mellifera*, the introduction of invasive pests and parasites is a continuing threat, as is the genetic dilution or extinction of locally adapted sub-species. Here we used metagenomic analyses of nineteen honey bee colonies from around Britain to compare host genetics,

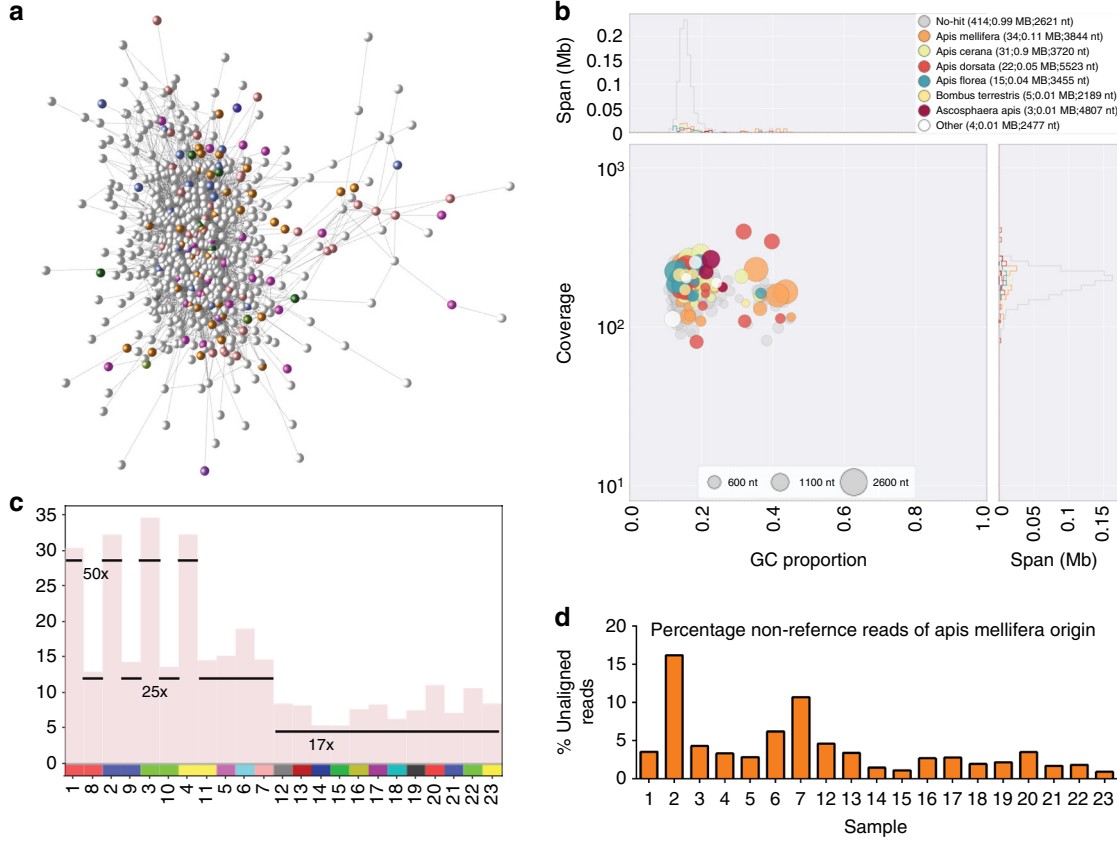

**Fig. 4** Putative *Apis mellifera* contigs. **a** A network component comprised of contigs which did not match the reference bee genome and were unassigned (white) or matched a non-reference species of bee (coloured). **b** Blobplot of these contigs (as in Fig. 2). **c** Mean base coverage per contig (*y*-axis) for each sample (*x*-axis) for the contigs in A. The sequencing depth (reference genome coverage) per sample is shown, showing that the number of reads mapping to these contigs is in direct proportion to the depth of sequencing. **d** A graph displaying the percentage of unaligned reads putatively identified as *Apis mellifera* from each sample

examine the complexity and connectedness of the bee microbiome, and quantify disease burden.

Using the reference honey bee genome and sequence data from 16 worker bees from each colony, we defined over five million SNVs with a relatively even distribution across all 16 chromosomes (Fig. 1). We also identified putative honey bee-derived sequences not represented in the reference C-lineage genome, likely because the reference is incomplete or because of genome variation between honey bee sub-species. The island of Colonsay in Scotland is a reserve for the northern European bee, *A. m. melifera*. Given the level of bee imports into Scotland, it was therefore reassuring – and perhaps surprising – to observe that the genotypes of other colonies from around Scotland were close to that of the Colonsay sample, although distinct from samples from *A. m. mellifera* breeding programmes in England. The low heterozygosity of Scottish *A. m. mellifera* and continued survival in face of imports may reflect natural selection for *A. m. mellifera* genotypes in the colder climates and shorter foraging season of northern Europe.

The whole organism-derived sequence data was also used to explore the composition of the communities of organisms living in or on honey bees. Non-*A. mellifera*-mapping reads were de novo assembled into contigs to generate 160 Mb of genomic sequence. Contigs were then assigned to species based on comparison to known genomes. A correlation network based on comparing the per-sample read coverage of these contigs (Fig. 2d) did not fully match the relatedness of the source bees (Fig. 1c), suggesting that both environmental and host genetic components drive microbiome composition. Our limited sampling (only

nineteen colonies) is not sufficient to unpick these interdependent drivers, but we note that samples from the Scottish coast, the central belt of Scotland and from England were grouped separately. These data are congruent with previous analyses of the roles of climate and forage in determining microbiome structure of honey bees[67,68].

In many animals, the gut microbiota form quasi-stable communities, with individual hosts harbouring somewhat predictable communities of different bacterial taxa. These different microbiome types have been associated with different gross physiological performance. In addition, changes in microbiota composition (dysbiosis) have been associated with the promotion of disease states in humans and other mammals[69,70]. Dysbiosis in honey bees may be an important correlate of bee and colony health[49,71–73].

In the honey bee gut, bacterial numbers are highest in the rectum, followed by the ileum, mid-gut and crop[71]. Lactobacilli are mainly found close to the rectum and, together with bifidobacteria, greatly outnumber other species[71]. We identified several contig clusters that likely represented single *Lactobacillus* species as well as a mixed-origin cluster (Fig. 4). Most of these were interlinked, revealing patterns of co-occurence of individual taxa. In contrast, *L. kunkeei*, an environmental cobiont reportedly indicative of poor health[71], formed a distinct, unlinked cluster. Samples 2 and 9 were technical replicates, and both had reduced diversity, containing only *G. apicola* and *Lactobacillus* species. The reason for this is unclear, but there was no evidence of pathogenic disruption of the sampled bees.

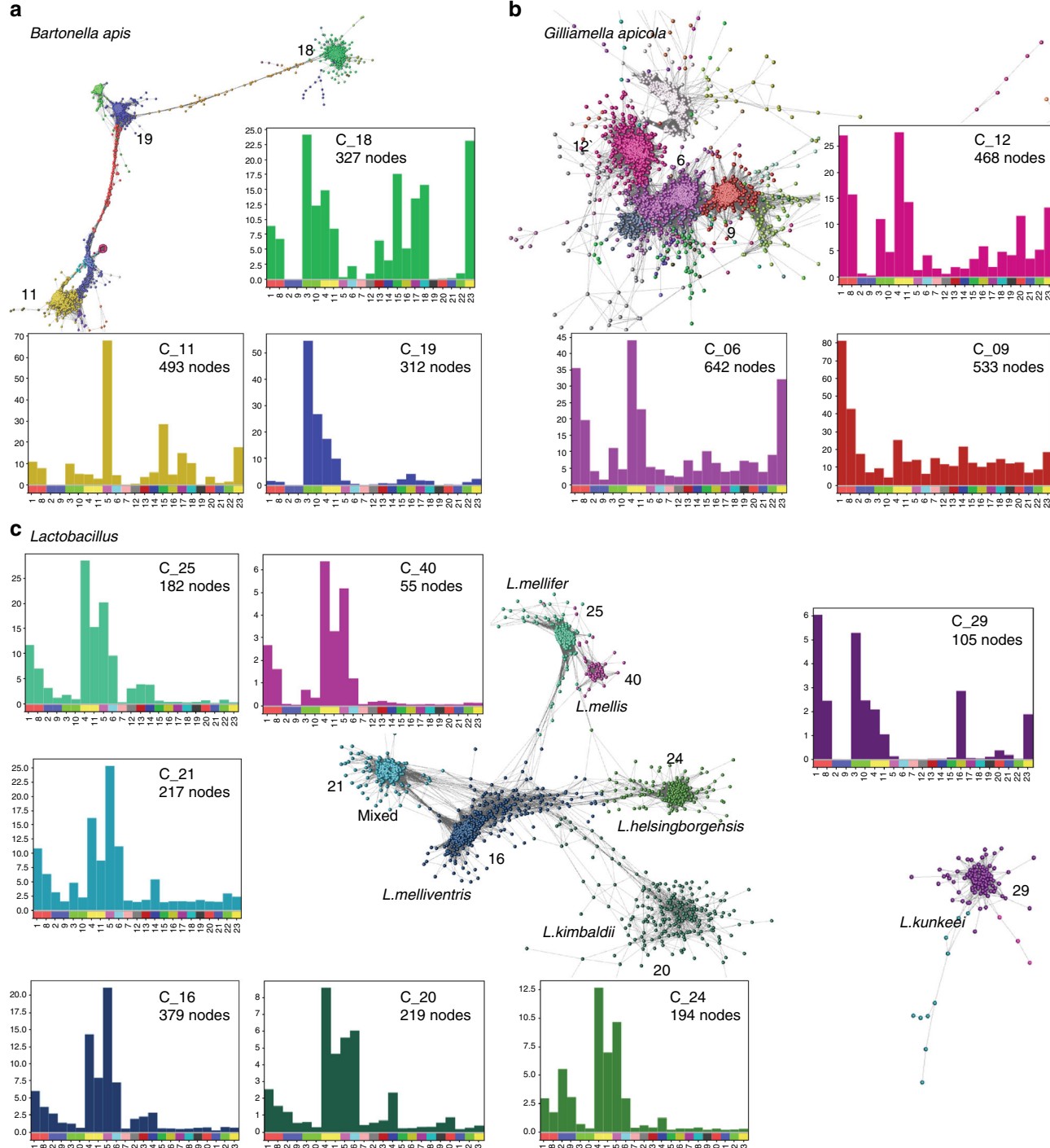

**Fig. 5** Communities of honey bee cobionts. Sub-networks of contig clusters from Fig. 3 coloured by cluster. Histograms show the mean base coverage per contig (*y*-axis) for each sample (*x*-axis). The number of contigs (nodes) in each cluster is also given. **a** *Bartonella apis*, **b** *Gilliamella apicola* and **c** several *Lactobacillus* species. Blobplots describing the taxonomy and cumulative span for each of these panels are presented in Supplementary Figure 3a–c

*Nosema* infection has been linked to immune suppression and oxidative stress of bee hosts[74]. Similarly *L. kunkeei* and *P. apium*, which are adapted to fluctuating oxygen levels predicted for the gut[75], have been associated with disease states in social bees, and negatively correlated with the amount of core commensal bacteria present[71]. The microbiome from sample 23 had a preponderance of reads mapping to the *L. kunkeei* cluster (Supplementary Fig. 3c), evidence of *P. apium* presence, much reduced representation of other *Lactobacillus* species, and the highest read

coverage of contigs associated with the pathogens *V. destructor*, AmFV and *A. apis*. Sample 23 may be an example of pathogen-induced dysbiosis, or of invasion by pathogens of a resident microbiome disturbed by other drivers. There was a high level of co-occurrence of different pathogens across samples, implying that colonies infected with one pathogen may be more susceptible to others. A meta-stable community may exist in the case of *Varroa destructor* and AmFV (Fig. 6c). However, we note a recent study reported identifying 0.5 Mb of sequence from *Varroa*

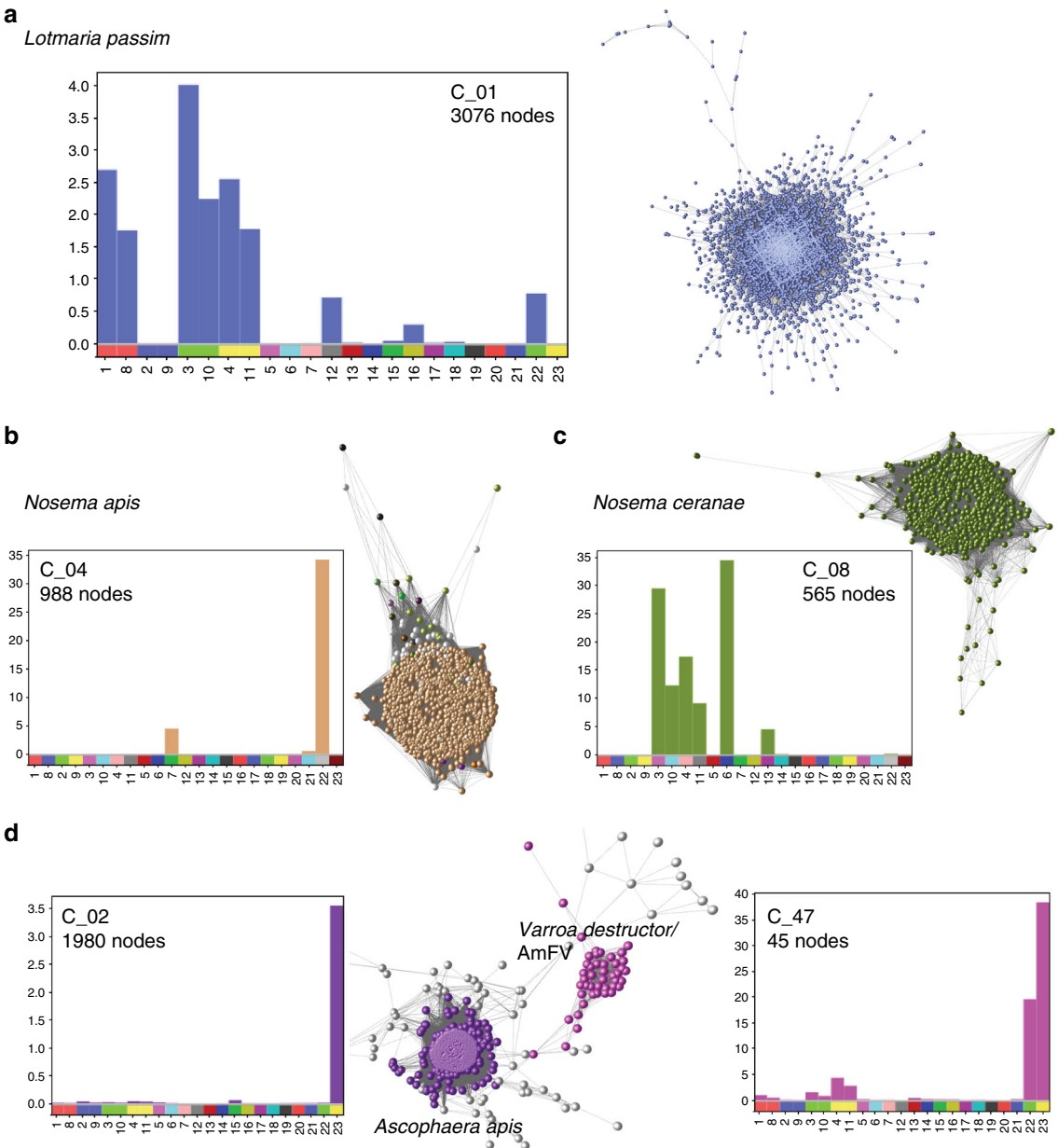

**Fig. 6** Disease associated components. Clusters associated with honey bee cobionts including mean base coverage per contig (*y*-axis) for each sample (*x*-axis). **a** *Lotmaria passim*, **b** *Nosema apis*, **c** *Nosema ceranae* and **d** a community of species including *Ascophaera apis* (associated with chalkbrood), *Varroa destructor* and *Apis mellifera filamentous virus* (AmFV). Blobplots describing the taxonomy and cumulative span for each panel are presented in Supplementary Figure 3d–J

reference genome to be of AmFV origin[76]. It is therefore possible that several of the contigs in our study matched with *Varroa destructor* are in fact of AmFV origin.

Several distinct contig clusters were assigned to *G. apicola* and *B. apis* suggesting the existence of genetically distinct subtypes of these highly prevalent bacteria. (Fig. 5a, b). *G. apicola* has a high diversity of accessory genes, associated with adaptation to different *A. mellifera* ecological niches[77,78]. Increased relative abundance of *G. apicola* has been associated with dysbiosis and host deficiencies[71]. Similarly, extreme displacement of *S. alvi* by *F. perrara* and *G. apicola* (and to a lesser extent by the opportunists *P. apium* and *L. kunkeei*) has been strongly associated with reduced bacterial biofilm function and host tissue disruption by scab-inducing *F. perrara*[49,73], leading to poor host development and early mortality. Blooms of *B. apis* have also been associated with poor health. This species exploits stressed, young, and old

bees, showing sporadic abundance in whole guts of newly emerged workers[58] and occurring uniformly across putatively dysbiotic foragers[56]. In support of this theory, samples from our study with the highest coverage of *G. apicola* and *B. apis* contigs also contained reads from pathogens such as *L. passim* or *Nosema* species. Significant positive correlation has been reported between infection levels of these parasites[79].

Our novel use of correlation networks (Fig. 3) to organise contigs based on their relative abundance across samples partitioned ~65% of contigs into clusters of sequences derived from an individual species and distinct micro-communities. Some sample-specific clusters, such as clusters 3 and 32, contained several core microbiome taxa. This may be a reflection of substrate specialisation based on host foraging[80]. However, several sample specific clusters contained contigs that had no informative taxonomic annotation, potentially revealing uncharacterised species. We

identified a cluster of unclassified contigs derived from a gregarine, with closest match to *Apicystis bombi*. The accuracy of our metagenomic analyses was confirmed by PCR and ribosomal DNA primers verified the species as *Apicystis bombi*. This is further evidence that managed honey bees can act as a reservoir for wild pollinator pathogens[65]; through increased understanding of honey bee molecular ecology and preventing disease transmission, we can indirectly improve wild pollinator health[81]. To our knowledge *Lotmaria passim* had not been previously identified in the UK. Its presence was confirmed for the first time in our study using the primers designed by Stevanovic et al.[82], further validating our sequencing inference. Other metagenomic binning approaches, such as MetaBAT, use both coverage information and sequence context (tetranucleotide frequencies) to bin genomes. Many parts of microbial genomes (e.g. 16 S/18 S cassettes, prophage, transposons, plasmids, AMR cassettes etc.) display different sequence composition than their host genome, but do show similar coverage patterns across multiple samples. For this reason, we wanted to avoid separation due to sequence composition, and therefore used only coverage in our network approach. We ran a MetaBAT pipeline and compared assemblies using CheckM which estimates completeness and contamination of bacterial genome assemblies based on the presence of unique genes[63]. On comparison, we found that MetaBAT results (Supplementary Table 4) were no better than those produced by our network approach (Supplementary Table 5). MetaBAT yielded eighteen bacterial genome assemblies at >80% complete compared to twenty assemblies using our method. However, assembly contamination levels (defined as % single copy genes seen more than once) ranged from 0–12% using MetaBAT compared to 0–18% seen using our method. Moreover, MetaBAT appeared to split certain eukaryotic clusters, e.g. the *Lotmaria passim* cluster (identified as *Leptomonas* by MetaBAT) was split into two bins and other clusters were missed entirely by MetaBat.

A whole-organism metagenomics approach has allowed us to describe the complexity of host-microbiome biology of British honey bees. Despite the limited size of our dataset and the incomplete genomic information for honey bee cobionts available to us, we have demonstrated the power of this approach using pooled samples in dual characterisation of the genotypic diversity of the honey bee, and the genomic diversity of its cobionts. Correlation networks are a powerful analytical approach that allowed us to cluster the sequence data to reveal interacting networks of bacterial and eukaryotic microbiota, in addition to classifying novel genomic sequences. As with the human and other animal microbiome projects, the precision of these analyses improves with additional data, permitting definition (and ultimately whole genome assembly) of novel genotypes of cobionts. To this end, the raw data from this project can be accessed through the Bee Microbiome Database, established and managed by the Bee Microbiome Consortium, a non-profit organisation of bee scientists for collecting, curating and analysing bee microbiome data[59]. While the sensitivity of metagenomic analyses is lower than that of PCR at present, complementation of cheap short-read data with low-coverage long-read data from isolated gut contents enhances the contiguity of assemblies and the functional inferences that can be derived them. This study highlights the potential to use this approach in routine screening, breeding programmes and horizon scanning for emerging pathogens.

## Methods

**Samples**. Nineteen samples of honey bee (each comprising sixteen workers collected from a single colony) were obtained from beekeepers in Scotland and England, with the help of Science and Advice for Scottish Agriculture (SASA) and Fera Science Ltd. The heads were not included in DNA extraction to avoid PCR

inhibitors present in the compound eyes of honey bees[83]. Wings and legs were not included as they were retained for wing morphometry and as a source for further DNA extraction. The thorax and abdomen of the sixteen bees from each colony were homogenised together in 2% CTAB buffer (100 mM Tris-HCl pH 8.0, 1.4 M NaCl, 20 mM EDTA pH 8.0, 2% hexadecyltrimethylammonium bromide, 0.2% 2-mercaptoethanol). Samples were incubated at 60 °C with proteinase K (54 ng/μl) for 16 h before incubating with RNaseA (2.7 ng/μl) at 37 °C for 1 h. After two chloroform:isoamyl alcohol (24:1) extractions, samples were ethanol precipitated, washed three times in 70% ethanol and resuspended in 0.1 TE. All genomic DNA samples were analysed for quantity (Qubit dsDNA HS Assay Kit, Thermo Fisher Scientific, Waltham, MA, USA), purity (Nanodrop, Thermo Fisher Scientific, Waltham, MA, USA) and quality (TapeStation, Agilent Technologies, Santa Clara, CA, USA).

**Sequencing**. All sequencing was performed by Edinburgh Genomics. DNAs were prepared for whole genome sequencing using the TruSeq DNA PCR-free gel free library kit (Illumina, Cambridge, UK) and, for eight samples, using the TruSeq DNA Nano gel free library kits (Illumina). For comparison, both types of libraries were prepared for four samples. 125 base paired-end sequencing was performed on an Illumina HiSeq 2500. Four samples were sequenced at 50× coverage, eight at 25X (including repeat sequencing of the four 50X samples) and 12 at 17X coverage. Data were screened for quality using FastQC v0.11.2 (Available online at: http://www.bioinformatics.babraham.ac.uk/projects/fastqc), and trimmed of low quality regions and adaptors using Trimmomatic v0.35[84] with parameters 'TRAILING:20 SLIDINGWINDOW:4:20 MINLEN:100.' These parameters remove bases from the end of a read if they are below a Phred score of 20, clip the read if the average Phred score within a 4 base sliding window advanced from the 5′ end falls below 20, and specify a minimum read length of 100 bases (the parameters used for all informatics analyses are also detailed in Supplementary Table 3).

**Variant calling on honey bee**. Reads were aligned to the reference *A. mellifera* genome, Amel_4.5 (INSDC assembly GCA_000002195.1) using BWA-MEM v0.7.8[85] with parameters -R and -M. Output files were merged and duplicates marked using Picard Tools v2.1.1 to create one BAM file per sample. This was filtered using SAMtools view v1.3[86] to retain only the highest confidence alignments using the parameters -q 20 (to remove alignments with a Phred score < 20) and -F 12 (to remove all reads that are not mapped and whose mate is not mapped).

Variants were called using GATK v3.5 in accordance with GATK best practice recommendations[87,88]. Local realignments were performed and base quality scores recalibrated using bee SNVs from dbSNP[89] build ID 140 (ftp://ftp.ncbi.nlm.nih.gov/snp/organisms/bee_7460/VCF/, downloaded 1 January 2016). GATK HaplotypeCaller was used with parameters emitRefConfidence, - GVCF variant index type – LINEAR, variant index parameter −128000, stand emit conf – 30, stand call conf - 30. The resulting VCFs, one per sample, were merged to create a single gVCF file using GATK GenotypeGVCFs to allow variants to be called on all samples simultaneously. Variant quality score recalibration was performed on this file using GATK VariantRecalibrator with parameters badLodCutoff – 3, -an QD, -an MQ, -an MQRankSum, -an ReadPosRankSum, -an FS, -an DP (specifying the above dbSNP data as both the truth set [prior = 15.0] and training set [prior = 12.0]). To identify any effect these variants may have upon protein-coding genes in the reference annotation, we used SNPeff v4.2[90]. A total of 5,302,201 variants were identified across the 19 samples.

**Population genetics analyses**. To give an initial overview of population structure, an Identity By State (IBS) analysis was performed using the R/Bioconductor package, SNPRelate[91]. Briefly, colonies were compared to the gVCF (see above) using autosomal and monomorphic SNPs only. The values of the resultant IBS matrix ranged from zero to one. Using this matrix, we constructed a network correlation graph for all of the samples, using the network analysis tool Graphia Professional (Kajeka Ltd., Edinburgh, UK), where each node represented a sample, and edges between nodes represented a correlation above the defined threshold between those samples (Fig. 1).

A more conservative approach was used to further examine the substructure of the population. SNVs were filtered using Plink v1.9[92]; again removing those not mapped to the autosomes, but also removing SNVs with a low genotyping call rate (<0.9), low minor allele frequency (<0.1), and pairwise linkage disequilibrium $r^2$>0.1 (for SNVs in 50 kb windows with a 10 kb step). The resulting 58,354 SNVs were submitted to unsupervised analyses in ADMIXTURE[93] for $1 \leq K \leq 5$ genetic backgrounds. To explore consequence of analysing genotypes from pooled DNA, individual genotypes simulated for 10 individuals per sampling location for each SNV were subjected to ADMIXTURE analysis. Briefly, for each SNV the allele frequency observed in a pooled sample was calculated from the read counts for each allele, and used to simulate ten genotypes assuming Hardy–Weinberg equilibrium. The efficacy of this process was tested using data from Harpur et al.[94], details of which are provided in the supplementary information (Supplementary Data 3). A distance matrix from the pooled DNA genotypes used in ADMIXTURE analyses was generated with Plink and analysed using the R package netview[54,55] (https://github.com/esteinig/netview), which analyses genetic structure using

mutual k-nearest neighbour (kNN) graphs. Graphs were created assuming $2 \leq k \leq 20$ nearest neighbours. The k-selection plot of these results together with the kNN $= 2$ network is presented in Supplementary Figure 1.

**Detecting regions of homozygosity**. We detected regions of homozygosity – from which can be inferred a reduction in selection strength relative to drift, or a recent selective sweep – using the pooled heterozygosity (Hp) method[95]. Sliding windows of 100 kb were advanced across each autosome with a step size of 50 kb. Within each window, we counted the number of reads corresponding to the most and least abundant SNP alleles (nmaj and nmin, respectively), then calculated $Hp = 2\Sigma nmaj \Sigma nmin/(\Sigma nmaj + \Sigma nmin)^2$. Only biallelic SNPs in the gVCF (see above) are included in this analysis. As certain genomic regions are harder to sequence at high depth, such as repetitive regions and areas of high GC content[96], we also controlled for per-site on-target read depth (considered a good predictor of variant detection sensitivity[97]) by restricting the analysis to those loci with a minimum read depth of 5 reads per locus per sample, i.e. accounting for regions under-covered for the purpose of variant detection (Fig. 1).

**De novo assembly and analysis of non-honey bee data**. De novo assembly was performed on all of the reads which did not map to the *Apis mellifera* reference genome using SPAdes v3.8.1[98]. The resulting contigs were filtered by length (>1 kb) and coverage (>2). BWA-MEM[85] was used to identify and remove reads mapping to these contigs and de novo assembly was performed on the remaining reads. This process was repeated for a total of five iterations. Input reads from each sample were mapped to each contig using BWA-MEM and base coverage/contig was calculated. Contigs with a cumulative base coverage from all samples less than half the SPAdes overall coverage were discarded. Using BLAST[99], contigs were compared to a set of custom databases: (1) HB_Bar_v2016.1[57]; (2) HB_Mop_v2016.1[57]; (3) nucleotide sequences of core microbiome species identified from literature[40,45,59,78]; 4. protein sequences of these species[40,45,59,78]; (5. NCBI nt[100]; 6. UniProt Reference Proteomes[101] using BLAST[99] and Diamond[102]. Files of all six sequence similarity searches were provided as input to BlobTools in the listed order under the tax-rule 'bestsumorder', i.e. a contig is assigned the NCBI taxid of the taxon providing the best scoring hits within a given file, as long as it has not been allocated a NCBI taxid in a previous file. BlobTools was used to visualise the coverage, GC% and best BLAST similarity match of the assembly, and to build a table of base coverage of contigs in each sample together with their taxonomic annotation. A network graph was constructed using *r* value of 0.99 comparing samples to each other based on correlations between their overall microbiome content, as well as contig coverage across the dataset (Fig. 2). This follows the approach used to compare gene expression values in transcriptomics data[103].

**Assessing genome completeness from metagenomic binning**. Using our assembled non-*Apis mellifera* contigs, we ran a metagenomic binning pipeline based on MetaBAT which uses both coverage information and sequence context (tetranucleotide frequencies) to bin genomes[63]. We then compared genome completeness from this analysis against our own using checkM (Supplementary Table 4). Because checkM is more appropriately applied to bacterial and archaeal genomes, we used BUSCO[64] to analyse our eukaryotic genome bins for *Ascophaera apis* (chalk brood), *Lotmaria passim*, *Nosema ceranae*, *Nosema apis* and *Varroa destructor* (Supplementary Figure 4).

**Primer design for identification of cobionts using PCR**. Custom primers were designed against the longest contigs we generated matching *Bartonella apis* (Bartonella_Fw 5′-CAGCAGCGCTTATTCCGTTC-3′, Bartonella_Rv 5′-AGTCAC-GAGCAACAATCGGT-3′) and the Gregarine species (Gregarine_F 5′-GACCACCGTCCTGCTGTTTA-3′, Gregarine_R 5′-GAGGTATCGGGTGC-CATGA-3′). Primers were run through NCBI BLAST to confirm specificity[99]. *Apicystis bombi* specific primers were used as described in Dias et al.[66]. Specific primers against *Nosema ceranae* were used as described by Chen et al.[104] and *Lotmaria passim* specific primers were used as described by Stevanovic et al.[82].

**Rarefaction analysis of microbiome sampling**. "Mean species richness" was calculated using the R package 'vegan'[105] for each sample at each of the sequencing depths used. Assembled contigs were analysed against the SILVA rDNA (16 S and 18 S) databases[106] instead of the NCBI nt database to assess species composition. Each contig identified as being from a unique species was counted as one "count" or incidence of discovering that species in the sample (Supplementary Figure 2).

## Data availability

Raw sequencing reads are freely available on the Short Read Archive (SRA) under BioProject ID PRJNA494922 (http://www.ncbi.nlm.nih.gov/bioproject/494922). A complete list of non-honey bee reference contigs and the BAM files indicating coverage of each contig from the 23 samples used in this study is freely available on Edinburgh DataShare. A table containing taxonomical annotation and the mean base coverage of each sample for each contig is also available here. This table was used to make the correlation network graph in Fig. 3 (http://dx.doi.org/10.7488/ds/2453). All scripts used are available in Supplementary Software or Github systems-immunology-roslin-institute/Honey-bee-metagenomics.

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

## Acknowledgements

The Bee Microbiome Consortium (http://wp.unil.ch/beebiome/consortium-members/). Dr. Jay Evans et al. for the HoloBee Mop dataset DOI: 10.15482/USDA.ADC/1255217 (https://data.nal.usda.gov/dataset/holobee-database-v20161). We thank Edinburgh Genomics for sequence generation. Science and Advice for Scottish Agriculture (SASA) and Fera Science Limited, (formerly the Food and Environment Research Agency, UK) for facilitating sample collection. Edward Carnell gave advice on map generation used in Fig. 1.

## Author contributions

T.R., M.W.B., F.H., M.B. and T.C.F. wrote the manuscript. T.R., M.W.B., D.R.L., S.J.B., D.W., M.W. and T.C.F. performed the analysis. M.W.B., F.H. and G.E.B. coordinated sample collection. B.D. and J.R.dM. provided the dataset from Jay Evans, and advice on microbiome analysis. M.B. and T.C.F. designed the project.

## Additional information

**Competing interests:** The authors declare no competing interests.

