## [Peer Review File · Nature Communications]

Reviewers' Comments:

Reviewer #1:

Remarks to the Author:

(expertise: metagenomics)

The manuscript by Ragan et al provides a metagenomic analysis of the microbial communities associated with the honey bee. By characterizing samples from 19 colonies, they reveal the presence of numerous known and unknown bee commensals and pathogens using a network analysis approach. While I think that the general experimental setup of the study is solid, the results of the provided network analysis remain somewhat superficial in my opinion.

Major point:

The obtained genomic clusters are described in general terms only, what is typically lacking is a clear genome-scale comparative analyses of the obtained data. The main reason for this is probably that the obtained genome clusters do not necessarily correspond to genomes (rather to collections of strains, or even related species or genera). Why haven't the authors performed a genome-resolved metagenomic binning approach to obtain draft-genomes rather than clusters? Apart from the fact that such draft-genomes would be a treasure trove for the bee microbiome community, it would allow for performing detailed comparative analyses between sampled colonies. I expect that interesting observations lie buried within such genome-level comparisons.

Minor points:

- Lines 146-148: Perhaps my math is off, but according to my calculations, assuming a bee genome of 236 Mbp, the equivalent coverage is quite a bit lower.
- The obtained genomic data of eukaryotic pathogens is interesting, as only very seldom are eukaryotic genomes reconstructed directly from metagenome data. How complete are these genomic assemblages?
- Suppl. Fig. 3, panel b: Should be "Nosema ceranae"
- Suppl. Fig. 3, legend: Should be "Bartonella apis"
- Suppl. Table 1: What can be inferred from the differences in % of reads that could be mapped back to the A.m. reference genome? I observe a rather large spread (77.1 – 97.3%) and wonder what the significance of this could be.
- Suppl. Table 2: It is unclear what 'Count' refers to – supposedly these are contigs, and not reads as indicated in the legend?
- Suppl. Table 2: Define 'No-hit' and 'Undef'. Could these potentially represent contigs from genomes of species that are being 'missed' with the current network approach (but could perhaps be salvaged via genome-resolved metagenomics)?

Reviewer #2:

Remarks to the Author:

The paper by Regan et al. uses NGS to study population genomics and the metagenome of 19 honeybee colonies mainly from Scotland. The study is comprehensive and very well written. The methods developed will be very useful and this study does indeed provide a useful baseline for honeybee metagenomic variation.

The main issue with this work, which I'm afraid is difficult to address post-hoc, is that it relies on opportunistic sampling and doesn't really address any hypothesis such as e.g. different management strategies or differences in environment, or indeed the different bee races or imported vs native bees that are addressed in the admixture section. This really limits what we can gain from this study beyond a very solid method and baseline description.

The discussion on diseases (l. 451 following) is interesting and certainly raises potential research questions, but is necessarily inconclusive.

other issues:

- Apicystis bombi: the description in the abstract makes it sound like as though you have found a new pathogenic gregarine, but as far as I can tell from the results and discussion, this is just Apicystis bombi, which has been described in honeybees before (however, it is not clear whether it is pathogenic in apis)

- l141: it would be good if you could expand in how far the colonies are representative of phenotypic diversity

-fig. 1: the figure would be improved by explaining what groups 1-4 are and what the colours mean e and f

- l. 240: an explanation of why the coverage was likely sufficient would improve the ms

- the correlation network analysis is very interesting but it could really do with more explanation in the results; did you take varying genome lengths into account for this? what could alternative explanations be and what does it really mean?

- I was a bit surprised to see Varroa reads, as the mites are quite big - were they not removed? Do you have data on the Varroa prevalence within the colonies, or any of the other diseases e.g. chalk brood?

-two wee typos l403 'was had', l411 'Varria'

Reviewer #3:

Remarks to the Author:

Regan and colleagues describe metagenomic data from 19 Apis mellifera colonies from the UK. Their analysis reveals interesting patterns in population structure and the diversity and occurrence of cobionts within the bee microbiome. These results are predominantly based on correlation networks that cluster contigs based on contig coverage and taxonomic annotation. I would strongly encourage the authors to open-source the code for their analyses and provide a means of reproducing and validating their pipeline (e.g. implementing the pipeline in Nextflow). The authors should consider using non-commercial software, particularly for the key components of their correlation networks, as

there are many alternatives freely available in the open-source ecosystem. Considering that the authors describe the analysis as a 'novel network analysis framework', reproducibility and validation are essential for the wider scientific community and the high standards of Nature Communications. The authors should also address limitations of their methods in the Discussion. Overall, their methods looks promising and the results of this paper are suitable for publication and of interest to the audience of the journal, if the authors can address the corrections and ensure that all analyses used for results and validation are cited appropriately and included in the Supplementary Materials.

References:

Line 47: Please use a peer-reviewed reference that more appropriately outlines the decline of pollinator populations globally than reference 2.

Line 55-56: Please support the statement "Trade in honey bees from different regions of the globe have unquestionably contributed to a rise in infectious diseases..." with appropriate peer-reviewed references. References 10 and 11 only refer to spread of disease between cross-pollinator transmissions.

Lines 73 -75: are repeating lines 55-56 and again only cite references 10 and 11, consider removing this redundant sentence entirely.

Lines 560: Please add the 'NetView' Molecular Ecology Resources citation in addition to reference 95, as requested by the developers of the R package (refer to their repository)

Line 187: reference 63 refers to a recent Nature paper by Rothschild et al. which on scanning does not appear to use kNN networks in any way. Please correct reference / cite the appropriate publications for 'NetView'

Spelling

Line 539: double period at the end of sentence.

Line 151: Figure 1 – bold font of 'Map' in sub-heading (F).

Introduction

Lines 102 -103: please clarify which host organism this refers to and provide a more appropriate reference – in this context it should be honey bees, but the citation refers to a paper discussing open questions in disease ecology and evolution in general.

Methods

General comment: please indicate for all bioinformatic analyses where they can be found as Figures or Supplementary Materials. While the detail of the methods is laudable (e.g. parameters for analyses), the link to results and discussions should also be made clear in the Methods.

General comment: considering the journal profile for this publication and the claim of a 'novel network analysis framework' the authors are strongly encouraged to implement their analysis into a reproducible pipeline framework (e.g. Nextflow) so that their analyses can be reproduced and validated. The authors must open-source their code on a suitable repository (e.g. GitHub, BitBucket) and are strongly encouraged to replace commercial software (i.e. Graphia Professional) with open-

source alternatives to enable the scientific community to replicate and validate their findings.

Lines 464 – 466: please clarify whether the individual worker extractions were pooled prior to sequencing each colony sample.

Lines 466 – 468: please clarify for a non-bee-expert audience why the wings, legs and head were removed for sequencing the metagenome, does this bias the microbiome composition?

Lines 516 – 522: this paragraph appears to describe a population genetic analysis based on the discovered SNVs. Please clarify why two different IBS matrices / distance matrices have been constructed with SNPRelate and Plink (including different filtering parameters, by the current description) and consider moving this component to the population genetic analysis paragraph. Please clarify why the SNVs for the correlation graph were not filtered for missingness, minor allele frequency, linkage disequilibrium as for Admixture.

Line 525: the authors assemble the non-bee metagenomics reads with Spade version 3.8.1 – there are several complications of metagenomics assembly that are addressed in metaSpades (Nurk et al. 2017) and compared by van der Walt et al. 2017 (10.1186/s12864-017-3918-9). Please clarify why Spades was used instead of metaSpades.

Lines 558 – 562: the authors use the kNN method to validate the correlation graph in Figure 1. As such it is critical to the results and the authors should provide the kNN networks in the Supplementary Materials. Please refer to the repository link for adding the correct citation for the implementation of 'NetView P' by Steinig et al. (2015) in Molecular Ecology Resources. Core methods of the R package are the same as the published package for Python according to developers (see 'Version' on GitHub) and citation is requested.

Results:

Line 151, Figure 1 component (A): four samples presumably corresponding to the duplicates (08 – 011) are missing from the map. However in component (C) the duplicates are identified as 01 – 04. Please clarify and adjust the map for correct sample enumeration.

Line 153, Figure 1 component (C): please clarify what the red edges in the correlation graph represent.

Line 177: well thought through, appreciate the validation for pooling samples and using Admixture.

Lines 186 – 188: I assume the k-NN network refers to 'NetView' – please clarify and link to relevant network graphs in Supplementary Materials.

Line 211: Please clarify why a correlation threshold of > 0.99 was used, is this an empirical foundation for this threshold?

Lines 327 – 328: proximity in the network visualization is strongly dependent on the graph visualization algorithm and one must be careful in its interpretation which should be based on connecting edges. I would suggest to remove 'In close proximity...' as it may mislead to think there is meaning in the proximity of the two clusters (which are barely connected by edges) – see also your point made in the Discussion Line 411 – 414.

Lines 337-338: this is a critical point that should also be mentioned in the Discussion. The authors are

(perhaps justifiably) excited about their method, but its limitations and in particular sensitivity should be discussed appropriately in the context of using metagenomics for 'routine screening, breeding programs and horizon scanning' in honey bees.

Discussion

Line 439: please discuss limitations of the method, in particular reduced sensitivity compared to PCR.

Reviewers' comments:

Reviewer #1

Major point: The obtained genomic clusters are described in general terms only, what is typically lacking is a clear genome-scale comparative analyses of the obtained data. The main reason for this is probably that the obtained genome clusters do not necessarily correspond to genomes (rather to collections of strains, or even related species or genera). Why haven't the authors performed a genome-resolved metagenomic binning approach to obtain draft-genomes rather than clusters? Apart from the fact that such draft-genomes would be a treasure trove for the bee microbiome community, it would allow for performing detailed comparative analyses between sampled colonies. I expect that interesting observations lie buried within such genome-level comparisons.

We thank the reviewer for pointing out this deficiency in the work. Metagenomic binning approaches, such as those used in MetaBAT and CONCOCT, use both coverage information and sequence context (e.g. tetranucleotide frequencies) to bin genomes. We have now run a metagenomic binning pipeline (based on MetaBAT, see <https://www.nature.com/articles/s41467-018-03317-6>) and we found that the MetaBAT results were no better than those produced by our network approach. Moreover, MetaBAT appeared to split certain eukaryotic clusters into two bins (identified as *Leptomonas* by MetaBAT, *Lotmaria passim* by our analysis), and we do not believe these are separate.

The pipeline clustered the contigs, and estimated the completeness and contamination of cluster-derived bacterial genome assemblies based on the presence of unique genes. 18 bacterial genomes were identified as >80% complete and with contamination levels, as defined as % single copy genes seen more than once, varied from 0 - 12%. Another 14 bacterial genomes were recorded as being between 4-77% complete. We have included the MetaBAT analyses of the same data as supplementary information **Supplementary Fig. 4** and **Supplementary Tables 4 and 5**].

It should be noted however that in our comparison of results, the network analysis seemed to perform better in clustering contigs. Firstly, MetaBAT only include just over 12k of 30k contigs in its analysis, thereby in some cases missing out significant clusters of contigs identified by the network analysis entirely. Moreover, MetaBAT appeared to split certain eukaryotic clusters. For example the largest cluster of contigs (identified as *Lotmaria passim* by our analysis, *Leptomonas* by MetaBAT) was split into two bins by MetaBAT and we do not believe these are separate.

Comparison of correlation network analysis and MetaBAT binning. Black arrow marks the cluster of *Lotmaria passim* contigs split into two by MetaBAT and red arrows point to some of the clusters identified by the network analysis ignored by MetaBAT.

It should also be noted that many parts of microbial genomes (e.g. 16S/18S cassettes, prophage, transposons, plasmids, AMR cassettes etc) display different sequence composition than their host genome, but do show similar coverage patterns across multiple samples. For this reason, we wanted to avoid separation due to sequence composition, and therefore used only coverage in our network approach. Finally, the objective of our paper is discuss the composition and variation of the honey bee metagenome in the UK; whilst we understand a detailed comparison of our approach to metagenomic binning techniques is of interest technically, we feel it to be beyond the scope of this paper.

These analyses have been added to the Results (pg. 21, ln 344-354), Discussion (pg. 26 ln 468-483), Methods (pg. 32 ln 627-635) and Supplementary Material (pgs. 40, 45).

Minor points:

- Lines 146-148: Perhaps my math is off, but according to my calculations, assuming a bee genome of 236 Mbp, the equivalent coverage is quite a bit lower.

Thank you for pointing out the error. This has now been corrected (pg. 6 ln 145).

- The obtained genomic data of eukaryotic pathogens is interesting, as only very seldom are eukaryotic genomes reconstructed directly from metagenome data. How complete are these genomic assemblages?

Suppl. Fig. 3, panel b: Should be "Nosema ceranae"

- Suppl. Fig. 3, legend: Should be "Bartonella apis"

Thank you for pointing out these typos, they have now been corrected (pg. 41).

- Suppl. Table 1: What can be inferred from the differences in % of reads that could be mapped back to the A.m. reference genome? I observe a rather large spread (77.1 – 97.3%) and wonder what the significance of this could be.

We agree that the spread in reads not mapping to the bee reference genome would appear to be large. This is in part down to one outlier sample (Sample 09) where the only 2.7% of reads did not map to the genome and at the other end of the spectrum another Sample 23, where nearly 23% of reads were non-mappers. We have wondered about this and postulated that in the first case, it could be that the samples taken were of young, recently emerged bees and the second outlier might be explained by the presence of a contaminant such as pollen contributing to the DNA. If one removes these two from the dataset, then the spread is rather less (82.2-93.7%) (pg. 42).

- Suppl. Table 2: It is unclear what 'Count' refers to – supposedly these are contigs, and not reads as indicated in the legend?

Thank you, this is correct. Count is referring to contigs rather than reads. The figure legend has been edited to reflect this (pg. 43).

- Suppl. Table 2: Define 'No-hit' and 'Undef'. Could these potentially represent contigs from genomes of species that are being 'missed' with the current network approach (but could perhaps be salvaged via genome-resolved metagenomics)?

A return of "No-hit" means that the contig did not match with anything in any database we searched against. "Undef" is a result from blast hits to sequences that do not have the taxonomic rank of "phylum" assigned to them (according to NCBI taxonomy). We have sought to make this clearer in a footnote of the table (pg. 43).

Reviewer #2:

The main issue with this work, which I'm afraid is difficult to address post-hoc, is that it relies on opportunistic sampling and doesn't really address any hypothesis such as e.g. different management strategies or differences in environment, or indeed the different bee races or imported vs native bees that are addressed in the admixture section. This really limits what we can gain from this study beyond a very solid method and baseline description.

We agree that these questions are impossible to address post-hoc and it was never the intention of the work to address the questions the reviewer highlights. While undoubtedly of significant interest, they require significant planning, resources and optimisation of the experimental design. The purpose of this work was to aid such studies in the future, by providing information on optimal depth of sequencing, an approach to mine these data and reference bee metagenome assemblies. This aim has been highlighted in the final paragraph of the discussion (pg. 26).

The discussion on diseases (l. 451 following) is interesting and certainly raises potential research questions, but is necessarily inconclusive.

other issues:

- Apicystis bombi: the description in the abstract makes it sound like as though you have found a new pathogenic gregarine, but as far as I can tell from the results and discussion, this is just Apicystis bombi, which has been described in honeybees before (however, it is not clear whether it is pathogenic in apis)

The reviewer is right to point out that we may have overstated the case for *A. bombi* being a pathogen. The abstract has been edited to reflect this (pg. 2 In 38).

- l141: it would be good if you could expand in how far the colonies are representative of phenotypic diversity

The colonies were chosen to reflect a broad spectrum of locations, with the aim of sampling a wide variety of microbiomes, and focused to a lesser extent the phenotype of the bees sampled. To this end, sampling sites favoured Scotland, where we had best contacts. However, because of the Scotland centric sampling, the samples are biased towards the European dark bee *Apis mellifera mellifera* (Henriques et al., 2018). A number of the samples from England also had a native bee bias, as they came from a dark bee breeding program. However, two samples, one of *Apis mellifera carnica* and the other of Buckfast, were specifically selected as controls for Southern bees and a

hybrid bee, respectively. We have attempted to clarify our approach in the manuscript (pg. 6 ln 138-143.)

-fig. 1: the figure would be improved by explaining what groups 1-4 are and what the colours mean e and f

The figure legend has been updated (pg. 7-8).

- l. 240: an explanation of why the coverage was likely sufficient would improve the ms

A line has been added here explaining the rarefaction analysis results (pg. 14 ln 246-250).

- the correlation network analysis is very interesting but it could really do with more explanation in the results; did you take varying genome lengths into account for this? what could alternative explanations be and what does it really mean?

The method and tools employed here, have been used widely to group/cluster genes based on their level of coexpression across samples, so called gene correlation networks (GCN). In this study the use of network analysis is based on the assumption that contigs originating from the same genome would have a similar relative distribution across samples. The same assumption is true for organisms whose presence is closely associated, i.e. they are likely cobionts forming micro-communities. In the current study, genome size was not taken into account. In many cases genome length is not known, although adjusting for genome length would still make little difference to the pattern of distribution and therefore would make little difference to the correlation matrix. To a certain degree the power of the approach increases with the number of samples analysed, as chance associations become more unlikely. However, even with the relatively few samples analysed here, a clear distinction between contigs from the same species was evident but interpretation of the meaning of associations between organisms should be treated with caution.

- I was a bit surprised to see Varroa reads, as the mites are quite big - were they not removed? Do you have data on the Varroa prevalence within the colonies, or any of the other diseases e.g. chalk brood?

We do not have figures for the prevalence of Varroa or indeed other diseases in the colonies sampled. The relatively large size of the mites means it should be possible to avoid knowingly sampling any bee with a Varroa mite present on its surface, the most likely explanation for their detection is that part of the mites, e.g. their legs or mouthparts may have remained on some of the bees sampled. They are also known to often hide between the abdominal sternites of bees.

-two wee typos l403 'was had', l411 'Varria'

Thank you for pointing out these typos, they have now been corrected.

Reviewer #3

Regan and colleagues describe metagenomic data from 19 Apis mellifera colonies from the UK. Their analysis reveals interesting patterns in population structure and the diversity and occurrence of cobionts within the bee microbiome. These results are predominantly based on correlation networks that cluster contigs based on contig coverage and taxonomic annotation. I would strongly encourage the authors to open-source the code for their analyses and provide a means of reproducing and validating their pipeline (e.g. implementing the pipeline in Nextflow).

We fully support the concepts of open science and for others to have the ability to access all data described and to be able to repeat these analyses. Therefore all scripts used are now provided in Supplementary File 1 while raw data is provided at datashare.is.ed.ac.uk

The authors should consider using non-commercial software, particularly for the key components of their correlation networks, as there are many alternatives freely available in the open-source ecosystem.

Others are welcome to use any network analysis platform they wish to perform such analysis, non-commercial or otherwise. In principle there is no reason others may not create a correlation matrix (this can be done in R, albeit slowly) and visualise the resulting graph in any tool of their choice. However, Graphia was used because it has been specifically designed for such analyses, rapidly calculating the correlation matrix, supporting the visualisation of massive graphs and clustering. Indeed, it is the only software which we have found capable of handling such a large graphs, at least within an acceptable time frame.

Considering that the authors describe the analysis as a 'novel network analysis framework', reproducibility and validation are essential for the wider scientific community and the high standards of Nature Communications. The authors should also address limitations of their methods in the Discussion.

Limitations include the sample size of the dataset and the incomplete genomic information currently available for honey bee cobionts. We have added some discussion on this point in the last paragraph of our discussion (pg. 26, ln 485-488).

Overall, their methods looks promising and the results of this paper are suitable for publication and of interest to the audience of the journal, if the authors can address the corrections and ensure that all analyses used for results and validation are cited appropriately and included in the Supplementary Materials.

References:

Line 47: Please use a peer-reviewed reference that more appropriately outlines the decline of pollinator populations globally than reference 2.

The sentence has been edited and a new reference has been cited outlining pollinator decline (ln 44-45).

Line 55-56: Please support the statement “Trade in honey bees from different regions of the globe have unquestionably contributed to a rise in infectious diseases...” with appropriate peer-reviewed references. References 10 and 11 only refer to spread of disease between cross-pollinator transmissions.

Thanks for highlighting this, a more appropriate citation has been added.

Lines 73 -75: are repeating lines 55-56 and again only cite references 10 and 11, consider removing this redundant sentence entirely.

Agreed, this has been done.

Lines 560: Please add the ‘NetView’ Molecular Ecology Resources citation in addition to reference 95, as requested by the developers of the R package (refer to their repository)

This reference has been added.

Line 187: reference 63 refers to a recent Nature paper by Rothschild et al. which on scanning does not appear to use kNN networks in any way. Please correct reference / cite the appropriate publications for ‘NetView’

These citations have been amended.

Spelling

Line 539: double period at the end of sentence.

Line 151: Figure 1 – bold font of ‘Map’ in sub-heading (F).

Thank you, these have been corrected.

Introduction

Lines 102 -103: please clarify which host organism this refers to and provide a more appropriate reference – in this context it should be honey bees, but the citation refers to a paper discussing open questions in disease ecology and evolution in general.

The ambiguity of host species reference has been removed with a specific example in bumblebees now added (pg. 4 ln 100-101).

Methods

General comment: please indicate for all bioinformatic analyses where they can be found as Figures or Supplementary Materials. While the detail of the methods is laudable (e.g. parameters for analyses), the link to results and discussions should also be made clear in the Methods.

Thank you, this is a good suggestion. The figure associated with each bioinformatics analysis has been indicated in Supplementary Table 3 (pg. 44-45) and throughout the text in the methods.

General comment: considering the journal profile for this publication and the claim of a 'novel network analysis framework' the authors are strongly encouraged to implement their analysis into a reproducible pipeline framework (e.g. Nextflow) so that their analyses can be reproduced and validated. The authors must open-source their code on a suitable repository (e.g. GitHub, BitBucket) and are strongly encouraged to replace commercial software (i.e. Graphia Professional) with open-source alternatives to enable the scientific community to replicate and validate their findings.

Agreed. Custom scripts have been provided in Supplementary File 1 to ensure full reproducibility. However, we have been unable to identify an open-source alternative program to Graphia which can handle a graph of this size (see above).

Lines 464 – 466: please clarify whether the individual worker extractions were pooled prior to sequencing each colony sample.

The pooling occurred prior to DNA extraction and sequencing. This has been made clear in the methods (pg. 28 ln 511-512).

Lines 466 – 468: please clarify for a non-bee-expert audience why the wings, legs and head were removed for sequencing the metagenome, does this bias the microbiome composition?

The head was not included in the extraction to avoid PCR inhibitors present in the compound eyes of honey bees (Boncristiani et al., *Apidologie* (2011) 42:457–460). Wings and legs were not included as

they were retained for wing morphometry and as a source for further DNA extraction (pg. 28 In 508-11).

Lines 516 – 522: this paragraph appears to describe a population genetic analysis based on the discovered SNVs. Please clarify why two different IBS matrices / distance matrices have been constructed with SNPRelate and Plink (including different filtering parameters, by the current description) and consider moving this component to the population genetic analysis paragraph. Please clarify why the SNVs for the correlation graph were not filtered for missingness, minor allele frequency, linkage disequilibrium as for Admixture.

We apologise for the ambiguity on this - the methods have been updated to explain the population genetic analyses in more detail. A new Supplementary Figure 1 has also been added including a NetView kNN network (pg. 33). While more conservative noise reduction was used to reveal the distinct sub-structures within the same graph (Fig. 1f), Fig. 1c was generated to provide an unbiased, complete overview of the population structure. Therefore the same conservative filtration was not applied.

Provided below (although not included in the manuscript) is a SNPRelate generated correlation graph including filtration for missingness and minor allele frequency (however, not linkage disequilibrium as this requires PLINK). The similarity in structure to the NetView network (Supplementary Figure 1 pg. 33) is unsurprisingly apparent. In the kNN network, Group 1 members tend towards the left with little, if any, C-lineage (Carniolan sample) composition. Group 2 members tend towards the right with more C-lineage.

Line 525: the authors assemble the non-bee metagenomics reads with Spade version 3.8.1 – there are several complications of metagenomics assembly that are addressed in metaSpades (Nurk et al.

2017) and compared by van der Walt et al. 2017 (10.1186/s12864-017-3918-9). Please clarify why Spades was used instead of metaSpades.

Thank you for highlighting this. metaSPAdes was not available when we began our analysis, but was incorporated as soon as it was available. Unfortunately we failed to update this in our methods. It has now been updated (Supplementary Table 3, pg 44-45).

Lines 558 – 562: the authors use the kNN method to validate the correlation graph in Figure 1. As such it is critical to the results and the authors should provide the kNN networks in the Supplementary Materials. Please refer to the repository link for adding the correct citation for the implementation of 'NetView P' by Steinig et al. (2015) in Molecular Ecology Resources. Core methods of the R package are the same as the published package for Python according to developers (see 'Version' on GitHub) and citation is requested.

Citations have been updated and a new Supplementary Figure 1 has been added including the k-selection plot and the k-NN network.

Results:

Line 151, Figure 1 component (A): four samples presumably corresponding to the duplicates (08 – 011) are missing from the map. However in component (C) the duplicates are identified as 01 – 04. Please clarify and adjust the map for correct sample enumeration.

Thank you, the figure has been updated to clarify the origin of duplicate samples 08 – 11 (pg. 7).

Line 153, Figure 1 component (C): please clarify what the red edges in the correlation graph represent.

The edges are coloured according to strength, ranging from thin and blue to thick and red. This has been added to the figure legend (pg. 7).

Line 177: well thought through, appreciate the validation for pooling samples and using Admixture.

Thank you.

Lines 186 – 188: I assume the k-NN network refers to 'NetView' – please clarify and link to relevant network graphs in Supplementary Materials.

Thank you for correctly identifying that. The text has been updated with proper reference to NetView and the Supplementary Figure 1 displaying the network (pg. 9, ln 191-193).

Line 211: Please clarify why a correlation threshold of > 0.99 was used, is this an empirical foundation for this threshold?

Essentially yes this was an empirically determined threshold. It was selected partly because at this threshold the graph was already very big (~20k nodes, 3.4 million edges) but mainly because at this threshold the graph had more distinct topology than at lower thresholds, where clusters begin to merge. The higher threshold also helps to minimise spurious associations (pg. 10 ln 217-221).

Lines 327 – 328: proximity in the network visualization is strongly dependent on the graph visualization algorithm and one must be careful in its interpretation which should be based on connecting edges. I would suggest to remove ‘In close proximity...’ as it may mislead to think there is meaning in the proximity of the two clusters (which are barely connected by edges) – see also your point made in the Discussion Line 411 – 414.

Agreed, the manuscript has been updated (ln 313-315 and 423-424).

Lines 337-338: this is a critical point that should also be mentioned in the Discussion. The authors are (perhaps justifiably) excited about their method, but its limitations and in particular sensitivity should be discussed appropriately in the context of using metagenomics for ‘routine screening, breeding programs and horizon scanning’ in honey bees.

Agreed, this point has now been reiterated in the last paragraph of the discussion (pg. 26-27).

Discussion

Line 439: please discuss limitations of the method, in particular reduced sensitivity compared to PCR.

This limitation has now been highlighted in the discussion (pg. 27, ln 497-498).

Reviewers' Comments:

Reviewer #1:

Remarks to the Author:

I am pleased to see that the authors have done a great effort in shaping up the manuscript, and have properly responded to any comments and inquiries. At this point I have no further issues or comments that need to be addressed. I was particularly happy to see that the authors have now included a metagenome binning analysis with Metabat (Metabat 2 I suppose?), and that they compare these results with their own network approach. I have to admit I am somewhat surprised that Metabat performed relatively poor on these datasets, although this could have many reasons (Metabat not optimized for eukaryotic genomes, and binning tools perform relatively poorly on small contigs, which might be a significant fraction of the data). It is good for the metagenomics/microbiome community (including myself) to know that there are other tools out there to pull out genomes from metagenome data. I am looking forward to see this manuscript published.

Reviewer #3:

Remarks to the Author:

Thank you kindly for addressing the points raised during review; the authors' changes improve on the manuscript and are satisfactory. I have no further issues with the latest version of the manuscript.